# From Synaptic Plasticity to Neurotoxicity: Endocannabinoid Influence on Addiction and Neurodegeneration

**DOI:** 10.3390/ijms262311632

**Published:** 2025-11-30

**Authors:** Balapal S. Basavarajappa, Shivakumar Subbanna

**Affiliations:** 1Center for Dementia Research, Nathan Kline Institute for Psychiatric Research, Orangeburg, NY 10962, USA; 2Department of Psychiatry, New York University Grossman School of Medicine, New York, NY 10016, USA

**Keywords:** cannabinoid receptors, alcohol, cell death, brain circuits, drugs of abuse, drug reinforcement, pathology

## Abstract

The endocannabinoid system (eCBS) is a versatile neuromodulatory network that orchestrates synaptic plasticity, reward processing, and neuronal homeostasis. Increasing evidence implicates eCBS dysregulation in both addiction and neurodegenerative (ND) disorders, suggesting overlapping molecular and cellular mechanisms underlying these conditions. This review synthesizes recent advances in understanding how eCBS components—cannabinoid receptors (CB1 and CB2), endogenous ligands (anandamide and 2-arachidonoylglycerol), and their metabolic enzymes—modulate dopaminergic and glutamatergic signaling within reward and reinforcement circuits. Chronic exposure to drugs of abuse, including alcohol, opioids, cocaine, and methamphetamine, perturbs eCBS homeostasis, promoting oxidative stress, neuroinflammation, excitotoxicity, mitochondrial dysfunction, and protein aggregation—pathological features common to Alzheimer’s, Parkinson’s, Huntington’s, and amyotrophic lateral sclerosis. These overlapping mechanisms disrupt neuronal integrity and contribute to progressive neurotoxicity, highlighting shared pathogenic pathways between addiction and neurodegeneration. Despite these advances, critical gaps remain in delineating how substance-induced eCBS alterations precipitate neurodegenerative cascades. Addressing these gaps will be essential for harnessing the eCBS as a therapeutic target to mitigate addiction-driven neurotoxicity and age-related cognitive decline.

## 1. Introduction

The endocannabinoid system (eCBS) is a multifaceted signaling network composed of endocannabinoids (eCBs)—lipid-derived messengers synthesized from long-chain polyunsaturated fatty acids—along with their cognate cannabinoid receptors and the enzymes that regulate their biosynthesis and degradation. Together, these components orchestrate a wide range of physiological and pathological processes across the central nervous system (CNS) and peripheral tissues [1,2]. The system was initially characterized as the neuronal substrate through which the principal psychoactive constituent of cannabis, Δ^9^-tetrahydrocannabinol (Δ^9^-THC), exerts its behavioral and physiological effects. The first breakthrough occurred with the pharmacological characterization of the cannabinoid receptor type 1 (CB1) in the rat brain by Devane and colleagues [3,4,5], followed by the cloning of the gene encoding this G protein-coupled receptor (GPCR) by Matsuda and collaborators [6,7]. These landmark discoveries laid the foundation for deciphering the molecular basis of cannabis-induced psychoactivity. These discoveries unfolded the way for defining the broader physiological roles of the eCBS [2,8,9,10,11,12,13]. Shortly thereafter, a second receptor, cannabinoid receptor type 2 (CB2), was identified in the rat spleen [14].

**Figure 1 ijms-26-11632-f001:**
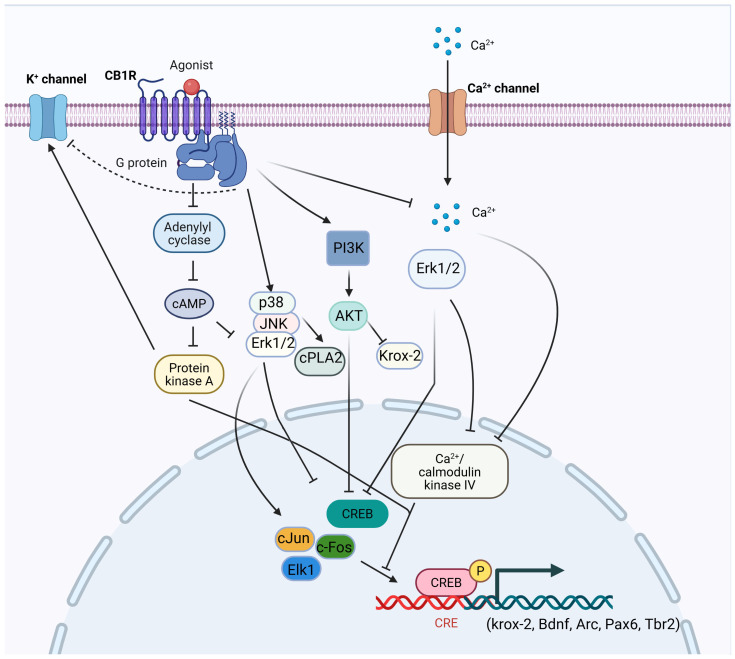
Schematic representation of CB1R signaling pathways. Both endogenous and mock cannabinoids exert their effects primarily through activation of CB1R [15,16,17,18]. CB1Rs are seven-transmembrane GPCRs embedded in the neuronal cell membrane [19]. Upon activation, CB1Rs inhibit AC activity [20,21,22], suppress N-type and P/Q-type voltage-gated calcium channels [23,24,25,26], and modulate potassium channels by inhibiting D-type and activating A-type as well as GIRKs [23,24,25,26,27]. CB1R signaling reduces presynaptic calcium influx through these mechanisms, thereby regulating neurotransmitter release and inhibiting synaptic transmission [28,29]. The Gα subunit primarily mediates the inhibitory actions on AC and calcium channels. At the same time, the Gβγ complex activates GIRKs and PI3K. Subsequent activation of the p38, JNK, and ERK1/2 pathways leads to phosphorylation of downstream effectors such as cPLA2, ELK-1, c-Fos, c-Jun, and CREB, promoting the transcription of target genes including *krox-24* and *BDNF* [16,30]. PI3K also mediates AKT-dependent inhibition of CREB activation [30]. Furthermore, CB1R-induced inhibition of AC decreases cAMP levels, thereby reducing protein kinase A (PKA) activity and the phosphorylation of specific K^+^ channels [18,31,32]. Under certain physiological conditions, CB1R signaling can also suppress ERK1/2 activation, leading to reduced CaMKIV and CREB phosphorylation, and consequent inhibition of *Arc* gene expression [33,34]. Stimulatory effects are indicated by arrows (→) and inhibitory effects by blunt-ended lines (⊥). CB1R, cannabinoid receptor 1; GPCRs, G protein-coupled receptors; AC, adenylate cyclase; GIRKs, G protein-coupled inwardly rectifying potassium channels; PI3K, phosphoinositide 3-kinase; p38, p38 mitogen-activated protein kinase; JNK, Jun N-terminal kinase; ERK, extracellular signal-regulated kinases, cPLA2, cytosolic phospholipase A2; Elk-1, E26 transformation-specific domain containing protein 1; c-Fos, an inducible transcription factor that regulates the delayed onset of effector genes; c-Jun, stress-activated protein; CREB, cAMP-response element binding protein; AKT, protein kinase B; cAMP, cyclic adenosine monophosphate; CaMKIV, calcium/calmodulin-dependent protein kinase IV; PKA protein kinase A. (Figure was created with BioRender.com).

CB1 is widely expressed throughout the CNS and peripheral organs [35,36,37]. In the brain, CB1 receptors are particularly enriched in the hippocampus, prefrontal cortex (PFC), basal ganglia, and cerebellum, with more moderate expression in the nucleus accumbens (NAc), thalamus, periaqueductal gray, and amygdala. Two major neuronal populations contribute to this distribution: gamma-aminobutyric acid (GABAergic) interneurons, which express high levels of CB1, and glutamatergic neurons, which express lower levels [38,39]. Additionally, CB1 receptors are present in monoaminergic nuclei such as the locus coeruleus (noradrenaline) and dorsal raphe nucleus (serotonin) [40,41], as well as in non-neuronal brain cells, including astrocytes, microglia, and oligodendrocytes [42,43]. Outside the CNS, CB1 receptors are expressed in the liver, gastrointestinal tract, skeletal muscle, reproductive tissues, adipocytes, immune cells, vascular tissues, and the cardiovascular system [44,45]. Activated CB1 receptors primarily couple to Gi/o proteins, leading to the inhibition of adenylyl cyclase and a reduction in intracellular cyclic adenosine monophosphate (cAMP) levels. Downstream signaling includes either activation [46,47,48,49,50,51,52] or inhibition [53,54] of mitogen-activated protein kinases (MAPKs) such as extracellular signal-regulated kinases (ERK), Jun N-terminal kinase (JNK), and p38, as well as modulation of cAMP-response element binding protein (CREB) phosphorylation, depending on cellular context (Figure 1). CB1 signaling also influences the phosphatidylinositol 3-kinase/ protein kinase B (PI3K/AKT) pathway, again in a context-dependent manner [31,55,56,57]. In neurons, CB1 activation suppresses voltage-gated calcium channels and enhances potassium channel conductance, thereby reducing intracellular Ca^2+^ and neurotransmitter release [51,58,59]. The released Gβγ subunits contribute to this effect by inhibiting N- and P/Q-type Ca^2+^ channels and activating A-type potassium channels [60,61]. In parallel, CB1-mediated Gβγ subunits initiate Src (non-receptor tyrosine kinase)-mediated cascades that activate MAPKs and focal adhesion kinases (FAK) [47,62].

Initially considered restricted to immune cells and peripheral tissues [35,63,64], CB2 receptors are now recognized as being expressed at low levels in the healthy CNS, particularly in microglia and astrocytes, with upregulation under pathological conditions such as brain injury, stroke, and neurodegeneration [65,66,67,68]. More recent studies have detected CB2 expression in neurons of the hippocampus, striatum, and thalamus [69,70], as well as dopaminergic neurons in the ventral tegmental area (VTA) [71]. Like CB1, CB2 couples predominantly to Gi/o proteins, inhibiting adenylyl cyclase and reducing cAMP levels (Figure 2) [35,72,73,74]. CB2 also activates the MAPK cascade, increasing ERK1/2 phosphorylation [63,73,75], and engages the PI3K–AKT pathway [76]. In addition, CB2 signaling can involve β-arrestin recruitment, which in some cases occurs independently of G protein-coupled receptor kinase (GRK) phosphorylation [77,78]. Ion channel modulation by CB2 includes inhibition of voltage-gated calcium channels (VGCC) and activation of GIRKs [79,80]. For detailed summaries of CB2 signaling, see recent reviews [81,82].

**Figure 2 ijms-26-11632-f002:**
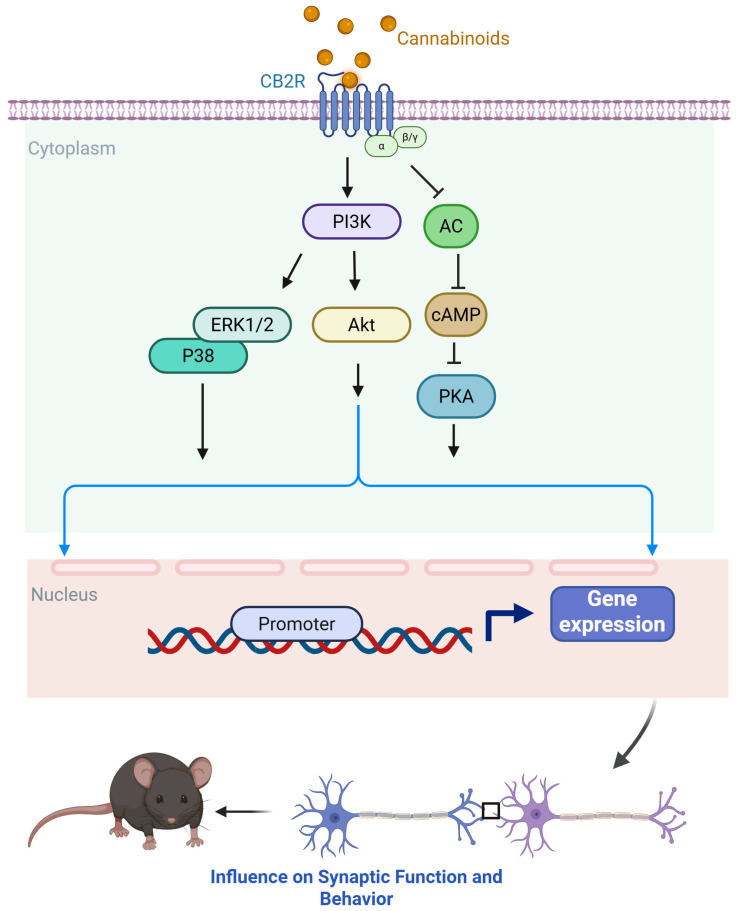
Schematic representation of CB2R signaling pathways. Both endogenous and synthetic cannabinoids interact with CB2R, a seven-transmembrane GPCR localized predominantly on immune cells, microglia, and, to a lesser extent, neurons [63,66,83,84]. Upon activation, CB2Rs couple primarily to Gi/o proteins, leading to inhibition of AC and a consequent reduction in intracellular cAMP levels. This attenuates PKA activity, thereby modulating downstream phosphorylation events that regulate cellular excitability, immune signaling, and gene transcription. CB2R activation also recruits multiple intracellular signaling cascades, including the MAPK pathways—ERK1/2 and p38 MAPK, and the PI3K/AKT pathway. These pathways contribute to cell survival, proliferation, and differentiation, and play critical roles in regulating neuroinflammation and neuroprotection. In immune and glial cells, CB2R-mediated signaling can modulate cytokine production, inhibit pro-inflammatory responses, and promote anti-inflammatory or restorative phenotypes. Additionally, CB2R activation has been linked to the de novo synthesis of ceramide, a bioactive lipid that serves as a second messenger in pathways governing apoptosis, oxidative stress responses, and cell cycle arrest [63,76,85,86,87,88,89,90,91,92]. CB2R signaling exerts broad regulatory effects on immune and neuronal function through these combined mechanisms. Stimulatory effects are indicated by arrows (→) and inhibitory effects by blunt-ended lines (⊥). Activation of CB2Rs signaling modulates presynaptic neurotransmitter release and influences synaptic function (dampens excitatory or inhibitory transmission), thereby shaping circuit activity and associated behavior. CB2R, Cannabinoid Receptor 2; GPCR, G protein-coupled receptor; AC, adenylate cyclase; PKA, protein kinase A; MAPK, mitogen-activated protein kinase; ERK, extracellular signal-regulated kinases; p38 MAPK, p38 mitogen-activated protein kinase; PI3K, phosphoinositide 3-kinase; AKT, protein kinase B (Figure was created with BioRender.com).

The cloning and characterization of CB1 also led to the identification of the first endogenous ligand, anandamide (AEA, N-arachidonoylethanolamine) (Figure 3) by the Mechoulam group in 1992 [5]. AEA only partially reproduced the psychotropic effects of Δ^9^-THC, leading to the subsequent discovery of a second major ligand, 2-arachidonoylglycerol (2-AG) [93,94] (Figure 3). Pharmacologically, AEA acts as a high-affinity partial agonist at CB1 with minimal CB2 activity, whereas 2-AG is a full agonist at both CB1 and CB2 but with lower binding affinity [95,96]. These distinctions highlight their complementary roles in regulating endocannabinoid tone. Beyond CB1/CB2, both AEA and 2-AG engage additional targets, including G-protein-coupled receptor 55 (GPR55) [96], transient receptor potential (TRP) channels such as TRPV1 [97], and nuclear receptors such as peroxisome proliferator-activated receptor alpha (PPARα) and gamma (PPARγ) [98], underscoring the pleiotropic nature of eCB signaling. Unlike classical neurotransmitters, eCBs are not stored in vesicles but are synthesized “on demand” in response to neuronal activity. Increases in intracellular calcium trigger their production and release from postsynaptic neurons, where they act as retrograde messengers at presynaptic CB1 receptors [99,100,101]. This mechanism modulates neurotransmitter release and can induce either short-term or long-term synaptic depression, depending on the circuit context [102]. AEA and 2-AG biosynthesis originates from membrane phospholipids. AEA is generated from N-acyl-phosphatidylethanolamine (NAPE) via NAPE-specific phospholipase D (NAPE-PLD), whereas 2-AG arises from diacylglycerol (DAG) through diacylglycerol lipase (DAGLα/β). Alternative enzymatic pathways for both ligands have also been described, adding further regulatory diversity [103,104]. Degradation is primarily hydrolytic: AEA by fatty acid amide hydrolase (FAAH), yielding arachidonic acid and ethanolamine, and 2-AG by monoacylglycerol lipase (MAGL), producing arachidonic acid and glycerol [105,106]. Both eCBs can also undergo oxidative metabolism via cyclooxygenase-2 (COX-2) and lipoxygenases, resulting in the generation of bioactive metabolites with distinct signaling properties [107]. The broad distribution of the eCBS across neural circuits and peripheral tissues underpins its involvement in diverse physiological processes, including cognition and memory, appetite regulation, motor control, pain modulation, immune responses, thermoregulation, sleep, stress adaptation, and reward processing [108]. In this review, we focus on recent advances in understanding the role of the eCBS in addiction and neurodegeneration, with particular emphasis on its dual significance as both a mediator of pathophysiology and a promising therapeutic target.

## 2. The Overview of the Reward and Reinforcement Circuit

The transition from recreational drug use to addiction represents a fundamental neurobiological shift in which voluntary consumption evolves into compulsive drug-seeking and taking, despite explicit awareness of harmful consequences. Progressive disruptions across multiple brain circuits drive this loss of control. Dysregulation within cortico-striatal and cortico-limbic networks alters reward sensitivity, incentive salience attribution, and associative conditioning, while impairments in prefrontal cortical regions compromise executive control, self-monitoring, and decision-making. In parallel, maladaptive changes in circuits regulating mood and interoceptive awareness exacerbate craving and heighten vulnerability to relapse. Collectively, these neuroadaptations provide a framework for understanding how chronic drug exposure reshapes brain function and entrenches the pathological state of addiction.

A central mechanism underlying this process is the mesolimbic dopamine system, which is consistently engaged by all well-studied drugs of abuse, including Δ^9^-THC. Classic studies demonstrated that such drugs elevate dopamine concentrations in terminal regions of this pathway [109,110]. The mesolimbic system originates in the ventral tegmental area (VTA; A10 dopamine neurons) and projects to limbic targets, most prominently the nucleus accumbens (NAc) [111]. Dopamine (DA) elevations within the NAc are tightly linked to the reinforcing properties of both natural rewards, such as food [112], and drugs of abuse [113], as well as to direct electrical stimulation of the medial forebrain bundle [114]. Significantly, this circuitry not only mediates the primary reinforcing effects of drugs and rewards but also supports associative learning. Environmental and contextual cues paired with drug use acquire secondary reinforcing properties [115,116,117], which can strongly precipitate relapse by eliciting craving and reinstating drug-seeking behavior [113,118]. Transient, phasic dopamine events encode both primary rewards and predictive cues [113,119], highlighting dopamine’s dual role in reinforcement and learning. Conversely, the negative affective state during drug withdrawal is associated with a downregulation of mesolimbic dopamine signaling, which further drives compulsive drug-seeking as a form of negative reinforcement [120,121,122].

### eCB Influence on Dopamine Transmission

Exogenous cannabinoids, such as Δ^9^-THC, reliably elevate extracellular DA levels in the ventral striatum [123,124,125]. This dopaminergic activation is mediated by CB1 receptor signaling, as pretreatment with the CB1 antagonist/inverse agonist SR141716A (rimonabant) abolishes Δ^9^-THC-induced increases in striatal DA [125]. Electrophysiological studies further demonstrate that cannabinoids enhance DA release in the NAc by increasing both the tonic firing rate and burst frequency of midbrain DA neurons [126], an effect reversed by CB1 blockade [127]. Beyond plant-derived cannabinoids, eCBs such as 2-AG can also promote DA neuron excitability, potentially through direct interactions with specific ion channels [128]. However, further in vivo studies are required to establish how this mechanism contributes to reward processing and reinforcement learning. A striking feature of cannabinoid action is that midbrain DA neurons themselves do not express CB1 receptors, implying that cannabinoids regulate DA activity indirectly through local circuit mechanisms. The VTA, the origin of mesolimbic DA projections, is composed of ~60% DA neurons, ~30% GABAergic neurons, and a smaller population of glutamatergic neurons (~3%) [129,130]. In addition to this intrinsic heterogeneity, the VTA receives extensive glutamatergic and GABAergic afferent inputs from limbic and sensory regions, many of which express CB1 receptors [131]. Thus, cannabinoid signaling at these presynaptic sites is well positioned to regulate DA neuron output dynamically.

One of the best-characterized mechanisms involves eCB-mediated modulation of GABAergic control over VTA DA neurons. In vitro application of the GABAA receptor antagonist bicuculline induces robust burst firing of VTA DA neurons, highlighting the powerful inhibitory influence of tonic GABA input on DA excitability [132]. Consistently, administration of the synthetic CB1/CB2 agonist WIN 55,212-2 decreases electrically evoked GABAA-mediated inhibitory postsynaptic currents (IPSCs) in VTA slices. This effect is prevented by CB1 antagonism [133]. These findings suggest that cannabinoids promote DA activity by reducing presynaptic GABAergic inhibition. Moreover, VTA DA neurons themselves can engage in activity-dependent “on-demand” synthesis and release of eCBs, which act retrogradely on CB1 receptors located on local GABA terminals. This feedback loop effectively disinhibits DA neurons, further amplifying phasic activity [131,134]. Cannabinoid modulation of DA signaling is not restricted to the VTA but also extends to its projection targets. In the NAc, CB1 receptors are expressed on both GABAergic and glutamatergic terminals, exerting bidirectional control over DA transmission. Activation of CB1 receptors on GABA terminals enhances DA release within the NAc and facilitates reward-related behaviors [135].

In addition to regulating DA neuron activity within the VTA, eCBs also act at VTA projection sites in the NAc. Within the NAc, eCBs interact with medium spiny neurons (MSNs) and prefrontal cortical glutamatergic inputs, particularly at prelimbic cortex–NAc synapses [136]. eCB signaling plays a multifaceted role in shaping dopamine transmission and reward-related activity, not by directly inhibiting DA release at D1-MSNs, but by modulating the glutamatergic and GABAergic afferents that regulate MSN excitability. Glutamatergic projections from the prefrontal cortex activate postsynaptic mGluR5 receptors, stimulating diacylglycerol lipase (DAGL) to produce 2-AG. The resulting 2-AG acts retrogradely on presynaptic CB1 receptors to reduce glutamate release, fine-tune excitatory drive onto D1-MSNs, and modulate excitatory transmission [137]. Disruption of this mGluR5–2-AG–CB1R pathway eliminates both natural and drug reward-seeking behaviors [138,139], and activation of these prefrontal inputs can induce 2-AG-dependent long-term depression (LTD) at NAc glutamatergic synapses [137,140,141]. Genetic studies further support this mechanism: conditional deletion of mGluR5 in D1- but not D2-MSNs abolishes 2-AG-dependent LTD and prevents reward-seeking behavior [142], whereas pharmacological enhancement of 2-AG in the NAc restores LTD and reward seeking [137].

Although eCB signaling often increases DA concentrations in the NAc by modulating inhibitory and excitatory inputs to midbrain DA neurons and their terminals, cannabinoid effects remain circuit- and cell-type-specific. Activation of CB1Rs on specific glutamatergic or cholinergic afferents can reduce NAc DA levels [143], underscoring the nuanced, pathway-dependent nature of cannabinoid action. Significantly, eCB signaling does not directly inhibit dopamine release from D1-MSNs—these neurons do not release dopamine—but instead shapes dopaminergic modulation indirectly by regulating presynaptic glutamate and GABA release, as well as forms of synaptic plasticity such as eCB-dependent LTD. Together, these findings extend the classical disinhibition model by demonstrating that eCB–glutamate interactions within the NAc are essential for balancing disinhibition and excitation across VTA–NAc circuits. Through this integrated regulation, both endogenous and exogenous cannabinoids exert robust control over reward processing, reinforcement, and vulnerability to addictive behaviors (Figure 4).

## 3. Drugs of Abuse and Alcohol-Induced eCBS Contribution to Neurodegenerative Disorders

Neurodegenerative (ND) disorders are among the leading causes of disability and mortality worldwide [56,152,153]. These conditions are characterized by the progressive loss of neurons, leading to impaired synaptic connectivity, neuronal death, and cognitive decline. Despite decades of intensive basic and clinical research, current therapies remain largely symptomatic rather than disease-modifying. This limitation likely reflects the slow and often asymptomatic progression of neurodegeneration, which can span decades before clinical symptoms become evident. Recent advances in molecular profiling and neuroimaging have enabled detailed mapping of signaling and transcriptomic alterations in postmortem human brains and in cellular and animal models of ND disorder. These studies have identified both classical pathogenic mechanisms and novel therapeutic targets, including synaptic degeneration—a hallmark shared across multiple ND conditions. Although large-scale genetic studies have revealed several causative mutations, a significant proportion of ND cases lack a clear genetic basis [154]. Increasing evidence implicates environmental factors—such as toxin exposure, nutritional deficiencies, psychosocial stress, and chronic alcohol or drug use—as major contributors to neurodegeneration and its associated behavioral and pathological outcomes. Despite extensive efforts, most pharmacological strategies have demonstrated limited efficacy [155,156], underscoring the urgent need for innovative therapeutic approaches capable of halting or reversing disease progression.

Over the past two decades, considerable progress has been made in elucidating the molecular underpinnings of ND disorders. Nonetheless, these disorders remain devastating and incurable, highlighting the necessity for novel and effective interventions. Emerging evidence positions the eCBS as a pivotal regulator of neuronal function, plasticity, and survival, thereby representing an attractive target for therapeutic development. Dysregulation of eCB signaling has been increasingly linked to the pathophysiology of major ND disorders, including Alzheimer’s disease (AD), Parkinson’s disease (PD), Huntington’s disease (HD), and amyotrophic lateral sclerosis (ALS) [56,157,158]. Moreover, chronic exposure to alcohol and other substances of abuse—such as methamphetamine, opioids, cannabis, and cocaine—induces neuropathological changes that resemble those seen in classical ND disorders [159,160,161,162,163,164]. These substances disrupt eCBS homeostasis, contributing to synaptic dysfunction, neuroinflammation, and neuronal loss that mirror key features of AD, PD, and related conditions. Building on these findings, this section of the review highlights emerging insights into how drug-induced alterations in the eCBS contribute to the onset and progression of cognitive decline and ND diseases, offering new perspectives for therapeutic intervention.

Heavy alcohol consumption remains a major global health concern associated with increased morbidity, mortality, and cognitive impairment. Its long-term impact on dementia-related neuropathology, however, remains unclear. Evidence suggests that alcohol abuse, particularly during adolescence, contributes to persistent behavioral and cognitive deficits. For instance, binge-like alcohol consumption impairs adult cognitive performance in rodent models, as demonstrated by deficits in the novel object recognition, Hebb–Williams, and Morris water maze tasks [165,166,167,168,169]. Chronic alcohol exposure during adolescence appears to be more detrimental than similar exposure during adulthood, producing greater long-term cognitive decline [170,171]. These findings suggest that both the pattern and developmental timing of alcohol use are critical determinants of its deleterious effects on adult and aging-related cognitive function. While some studies report that moderate alcohol consumption is associated with better cognitive performance [172,173,174] or no clear relationship [175,176], heavy drinking is consistently linked to cognitive decline [177,178]. Structural neuroimaging studies have yielded mixed findings: moderate alcohol use has been associated with β-amyloid (Aβ) deposition in middle-aged and older adults without alcohol use disorder (AUD) or dementia [179], whereas other studies found no significant differences in Aβ levels among individuals with AUD [180]. Epidemiological data indicate that older adults (>65 years) who consume more than 14 alcoholic drinks per week have an increased risk of AD and all-cause dementia compared with those who drink less [164,181]. Heavy drinking—defined as two or more drinks per day—has also been associated with earlier disease onset, by up to four years, compared to lighter (less than two/day) drinkers [182]. Future research employing diverse animal models, beyond transgenic systems, is crucial to better understand how alcohol exposure across developmental stages influences aging and AD pathology.

Evidence for a direct relationship between alcohol consumption and PD remains inconsistent. Some studies have reported gender differences in risk, with a higher relative risk in men [183]. In contrast, others suggest that chronic alcohol intake increases PD susceptibility [184]. In alcohol-preferring rats, reduced DA overflow and increased α-synuclein (αSYN) accumulation in the nucleus accumbens have been observed, suggesting that αSYN-mediated alterations in dopaminergic neurotransmission may contribute to alcohol-related motor impairments [185]. However, a definitive association between alcohol use and PD risk has yet to be established, warranting further investigation. Similarly, studies on alcohol and ALS have produced inconsistent results, with population-based analyses suggesting no significant effect of alcohol consumption on ALS risk [186]. Evidence for alcohol influence on the development and progression of HD is inadequate, with only anecdotal reports describing delayed HD diagnosis in alcoholic individuals with a family history of the disease [187].

### 3.1. Alcohol Abuse-Induced eCBS Alterations and Neurodegeneration

Extensive research in rodent models has demonstrated that various forms of alcohol abuse lead to structural and functional deficits in brain regions critical for cognition [147,148,149,150], along with significant alterations in the eCBS [188,189,190,191]. However, only a limited number of studies have examined the relationship between alcohol-induced eCBS dysregulation and ND-related pathology [188,189,191,192,193,194,195,196]. For example, inhibition of FAAH using URB597 attenuates the effects of adolescent alcohol exposure on memory, neuroinflammation, and brain-derived neurotrophic factor (BDNF) levels in adulthood [194]. Likewise, activation of CB2 receptors prevents alcohol-induced impairment of neurogenesis [193], while CB1 receptor blockade or genetic deletion mitigates adult cognitive and transcriptional deficits induced by early alcohol exposure [54,197,198,199,200,201]. Recent findings from a transgenic AD mouse model amyloid protein precursor/presenilin (APP/PSEN) show that adolescent binge-like alcohol exposure accelerates cognitive decline at 6 and 12 months, accompanied by increased Aβ42 deposition and reduced expression of the 2-AG-metabolizing enzyme MAGL [202]. Alcohol exposure also increased CB2 and DAGLα expression in wild-type, but not AD, mice. These studies collectively suggest that alcohol-induced eCBS dysregulation contributes to age-related cognitive decline and dementia-like pathology. Targeting specific eCBS components may therefore offer a promising strategy to prevent or delay AD progression. For instance, FAAH inhibition not only reduces alcohol-related behaviors [189] but also attenuates AD pathology and disease progression in mouse models [203,204,205]. Furthermore, tau accumulation enhances CB2 expression in neurons, contributing to neurodegeneration in a tauopathy model (hTAUP301L) [206]. Given the overlapping hippocampal pathology observed in aging, alterations in the eCBS, and AUD, it is essential to elucidate how alcohol-induced neurobiological changes at distinct developmental stages predispose the brain to pathological aging and dementia. Although the roles of AUD [207,208,209] and eCBS [210,211,212,213] in driving epigenetic alterations have been investigated, a comprehensive analysis integrating AUD-induced eCBS gene expression, chromatin remodeling, and protein-level changes across specific brain regions and cell types is still lacking. Such studies are crucial to understanding how epigenetic mechanisms shape cellular phenotypes in addiction and contribute to neurodegenerative disorders.

### 3.2. Drugs of Abuse-Induced eCBS Alterations and Neurodegeneration

#### 3.2.1. Cocaine

Drugs of abuse exert profound neurotoxic effects that compromise brain health and contribute to various ND disorders. Chronic cocaine exposure, for instance, promotes neuroinflammatory processes through oxidative stress and mitochondrial and endoplasmic reticulum (ER) dysfunction, accompanied by aberrant activation of microglia and astrogliosis. These events collectively trigger excitotoxicity and neuronal injury [214,215]. Cocaine-induced ER stress, autophagy, and inflammatory signaling lead to protein misfolding and aggregation reminiscent of those observed in ND pathologies [216,217,218]. Moreover, cocaine stimulates astrocyte proliferation and neuroinflammation, further exacerbating neurodegenerative cascades [219].

Cocaine exposure also induces hyperphosphorylation of tau protein in the CNS—an event central to AD pathogenesis [220]. The mechanisms underlying cocaine neurotoxicity are multifactorial and involve excessive cytosolic and synaptic DA accumulation, mitochondrial impairment, oxidative stress, neuroinflammation, pro-apoptotic signaling, altered neuronal plasticity, and disrupted neurogenesis. Notably, cocaine damages dopaminergic neurons and increases the risk of motor dysfunction, showing a threefold higher incidence of dopamine-related motor symptoms among users [221,222,223]. α-Syn directly interacts with and functionally couples to the dopamine transporter (DAT), enhancing DA uptake and accelerating DA-induced apoptosis [224]. Elevated α-syn levels have been observed in the brains of cocaine users, suggesting that increased α-syn production may contribute to PD progression [225]. Consistent with this, Parkinsonian-like motor anomalies have been detected in individuals with a history of cocaine abuse, even after prolonged abstinence [226]. Cocaine abuse has also been associated with aggravated ALS symptoms, suggesting a potential role in disease initiation or acceleration [227]. Although both cocaine [228] and ALS [229] are linked to glutamate-mediated excitotoxicity, the mechanistic connection remains poorly defined and warrants further investigation.

#### 3.2.2. Amphetamine

Amphetamine-type stimulants (e.g., methamphetamine, MA) likewise induce oxidative stress, apoptosis, and dopaminergic neurodegeneration, leading to behavioral and cognitive impairments [230]. Substantial evidence indicates that MA toxicity drives degeneration of dopaminergic neurons within the nigrostriatal pathway, a hallmark shared with ND disorders such as AD and PD [231,232,233,234]. MA exposure increases the expression of AD-related proteins, including phosphorylated tau and APP [235], and chronic MA users exhibit a heightened risk of PD [236]. Mechanistically, MA induces PD-like pathology via epigenetic upregulation of α-syn expression [237] and by promoting nigrostriatal toxicity through depletion of superoxide dismutase 1 (SOD1) and heightened oxidative stress [193]. Notably, oxidative damage due to SOD1 mutations is also implicated in ALS pathogenesis [193]. Collectively, these observations indicate that, although current studies remain limited, drugs of abuse pose a significant risk for the development and progression of various ND disorders. This highlights the urgent need for systematic investigations to elucidate the mechanistic links between substance abuse and neurodegeneration—two major contributors to morbidity and mortality in the aging population.

Despite the substantial burden imposed by drugs of abuse on the CNS, current pharmacotherapies remain largely ineffective, partly due to an incomplete understanding of the underlying neurobiological mechanisms. The eCBS has emerged as a critical modulator of the neurophysiological and behavioral effects of addictive substances [194,195]. Conversely, drugs of abuse profoundly impact eCBS signaling [107], and eCBS dysregulation has been reported during aging [116] and across multiple ND disorders [110].

Although direct evidence linking drug abuse, eCBS alterations, and ND is limited, several findings highlight eCBS as a promising therapeutic target. For instance, inhibition of FAAH significantly reduces cocaine-induced modulation of CB1 receptor activity [238]. It may confer protection against cocaine-related neurotoxicity [197]. CB1 receptor antagonists attenuate cocaine-induced cellular adaptations, excitatory synaptic alterations, and memory reconsolidation in rodents [239,240,241], processes critical for cognitive and addictive behaviors. Genetic variants in CNR1 (the CB1 receptor gene) and the FAAH rs324420 polymorphism have been associated with MA abuse. MA exposure also alters eCBS enzyme expression, increasing NAPE-PLD while reducing CB1 receptor and FAAH levels [200], and eCBs mediate MA-induced synaptic activity [201]. The eCBS exerts homeostatic control over neuronal integrity and function throughout aging [110,116,202]. Furthermore, cannabinoids display broad neuroprotective properties against oxidative stress, excitotoxicity, glial reactivity, and protein aggregation—all hallmarks of ND pathology. Moreover, eCBS activity supports adult neurogenesis, a process disrupted in many NDs with prolonged prodromal phases. The localization of eCBS components within key CNS structures that govern neuronal survival and plasticity underscores its potential as an intrinsic defense system against neurotoxic insults, including oxidative stress, protein dysregulation, and inflammation. Experimental evidence supports the neuroprotective capacity of both endogenous cannabinoids and phytocannabinoids (for review, see [202,203,204]). CB2 receptor agonists, in particular, exhibit protective effects against drug-induced neurotoxicity [203] and demonstrate neuroprotection in animal models of AD [204], PD [117,205], HD [206,207], ALS [208,209], and other NDs [110,210]. Although the optimal eCBS targets likely vary by disease type and progression stage, continued investigation into the interplay between drug abuse, eCBS modulation, and neurodegeneration will be essential for the development of targeted neuroprotective therapies.

In addition, given the converging hippocampal pathologies associated with aging, eCBS dysfunction, and exposure to drugs of abuse, it is critical to evaluate how drug-induced neurobiological changes at different developmental stages heighten susceptibility to pathological aging and dementia. While previous studies have explored the effects of drugs of abuse [242,243,244,245,246] and eCBS modulation [212,213,214,215] on epigenetic regulation, a comprehensive investigation integrating drug-induced alterations in eCBS gene expression, chromatin remodeling, and proteomic landscapes across specific brain regions and cell types is still lacking. Such multidimensional analyses are essential to clarify how epigenetic mechanisms govern cellular plasticity in addiction and drive the progression of neurodegenerative disorders.

## 4. Conclusions

The eCBS occupies a pivotal intersection between synaptic plasticity, addiction, and neurodegeneration. Drugs of abuse and alcohol profoundly disrupt eCB signaling, triggering oxidative, inflammatory, and proteostatic imbalances that accelerate neuronal dysfunction and death. Conversely, the eCBS exerts endogenous neuroprotective actions by regulating neurotransmission, neurogenesis, and glial activity. Evidence from preclinical models highlights that modulating specific eCBS targets—such as CB1, CB2, FAAH, and MAGL—can attenuate drug-induced neurotoxicity and ameliorate neuropathological features of Alzheimer’s, Parkinson’s, Huntington’s, and motor neuron diseases (Figure 5). However, translational studies remain limited. Future research integrating molecular, imaging, and behavioral approaches is essential to delineate the causal pathways linking drug abuse, eCBS dysregulation, and neurodegeneration. Such efforts will advance mechanistic understanding and foster the development of cannabinoid-based or eCBS-targeted interventions to restore neuronal resilience and prevent neurodegenerative decline across the lifespan.

## Figures and Tables

**Figure 3 ijms-26-11632-f003:**
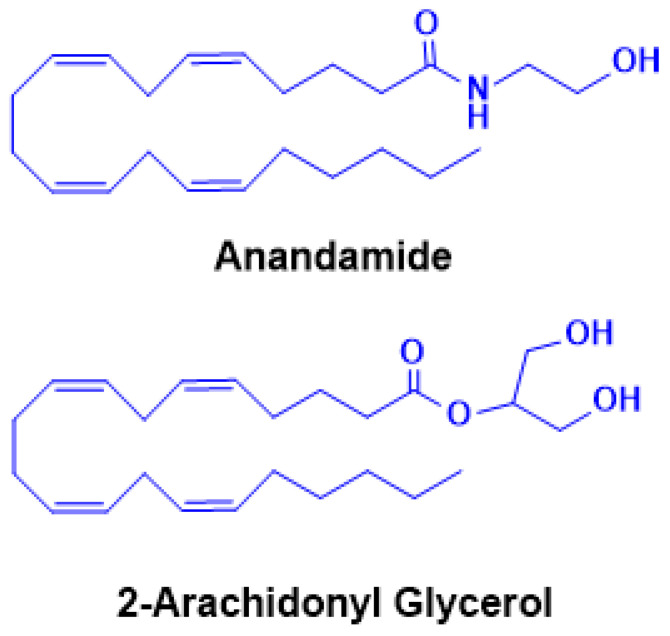
Chemical structures of endogenous cannabinoids anandamide (AEA) and 2-arachidonoylglycerol (2-AG). The two principal endocannabinoids, AEA and 2-AG, are lipid-derived signaling molecules synthesized on demand from membrane phospholipid precursors. Both AEA and 2-AG act as endogenous ligands for cannabinoid receptors CB1R and CB2R, mediating a wide range of physiological and neurobiological processes.

**Figure 4 ijms-26-11632-f004:**
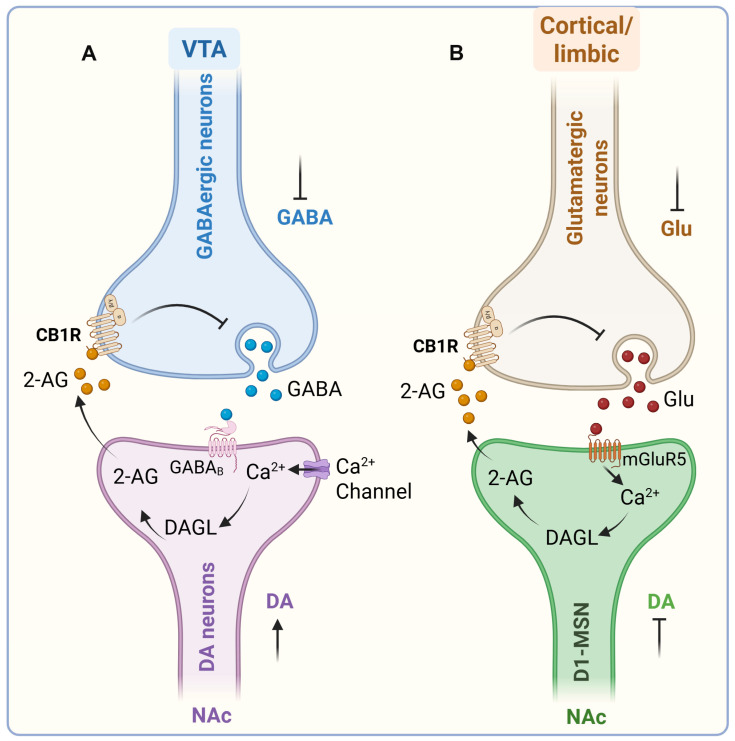
Endocannabinoid-dependent mechanisms mediating natural reward and drug-seeking behaviors. Two major endocannabinoid (eCB)-dependent mechanisms within the mesocorticolimbic circuit contribute to both natural and drug-induced reward processing. (**A**) In VTA, eCB signaling regulates dopaminergic activity via disinhibition. Under basal conditions, DA neurons are tonically inhibited by GABAergic interneurons through GABA_B_ receptor activation. Presentation of reward-predictive or drug-associated cues induces phasic DA firing, elevating intracellular Ca^2+^ and activating DAGL. DAGL catalyzes 2-AG synthesis, which acts retrogradely on presynaptic CB1Rs on GABA terminals, suppressing GABA release and disinhibiting DA neurons. This promotes burst firing and enhanced DA release [144,145,146]. Blockade of GABA_B_ or CB1Rs prevents this reward-seeking response, confirming CB1R-dependent disinhibition as a key modulatory mechanism. (**B**) In the NAc, eCB–glutamate interactions fine-tune excitatory drive onto D1-MSNs. Glutamatergic inputs from the prefrontal cortex activate mGluR5, stimulating DAGL and 2-AG synthesis. The released 2-AG retrogradely activates CB1Rs on glutamatergic terminals, inhibiting further glutamate release and modulating excitatory transmission [137]. Inhibition of either mGluR5 or CB1Rs abolishes both natural and drug reward-seeking behaviors [138,139]. Reward-related glutamatergic afferents promote DA neuron burst firing via iGluR activation, balanced by GABAergic inhibition [147]. Reduced GABAergic tone disinhibits DA neurons, increasing DA release onto MSNs. Ca^2+^ influx through voltage-gated channels, together with mGluR1/5 activation, drives “on-demand” production of 2-AG (and to a lesser extent, AEA). Retrograde 2-AG signaling through CB1Rs on GABAergic and glutamatergic terminals enhances phasic DA release and plasticity [114]. CB1Rs are denser on GABAergic than glutamatergic terminals, making cannabinoids (e.g., Δ^9^-THC, 2-AG) full agonists at GABA sites but partial agonists at glutamatergic terminals [148,149,150]. The role of AEA remains less defined due to asymmetrical localization of its biosynthetic (NAPE-PLD) and degradative (FAAH) enzymes. CB1Rs on local GABAergic and glutamatergic terminals further regulate accumbal DA output—enhancing it via GABA inhibition or reducing it through glutamate suppression [151]. eCB, endocannabinoid; VTA, ventral tegmental area; GABA_B_, gamma-aminobutyric acid B; DAGL, diacylglycerol lipase; 2-AG, 2-arachidonoylglycerol; CB1Rs, cannabinoid type 1 receptors; NAc, nucleus accumbens; D1-MSNs, D1 medium spiny neurons; mGluR5, metabotropic glutamate receptor 5; iGluR, ionotropic glutamate receptor; NAPE, N-acyl-phosphatidylethanolamine; PLD, phospholipase D; FAAH, fatty acid amide hydrolase; Δ^9^-THC, tetrahydrocannabinol; GABA, γ-aminobutyric acid; ACh, acetylcholine. ↑, increase; ⊥, decrease (Figure was created with BioRender.com).

**Figure 5 ijms-26-11632-f005:**
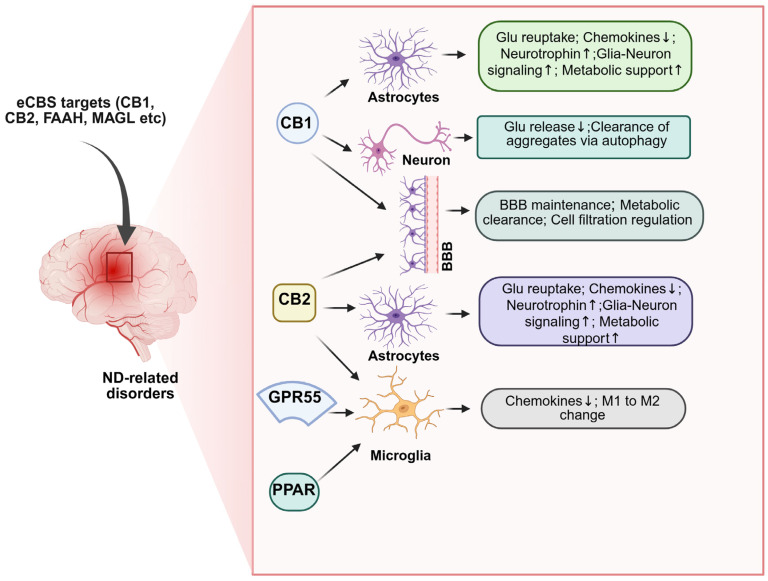
Endocannabinoid system-mediated neuroprotection in neurodegenerative disorders. The eCBS provides neuroprotection by coordinating actions across neurons, astrocytes, microglia, and the vasculature. Neuronal protection involves CB1 receptor-mediated inhibition of excessive glutamate release at presynaptic terminals, modulation of postsynaptic calcium channels, and maintenance of glutamate/GABA balance, limiting excitotoxicity [56,247,248,249]. CB1 signaling also promotes autophagy, facilitating clearance of damaged proteins and aggregates [250,251,252]. Glial-mediated protection occurs via CB1, CB2, GPCRs, and PPAR receptor activation. In astrocytes, eCBs signaling supports energy metabolism, glutamate clearance, neurotrophin production, and the release of anti-inflammatory factors, enhancing neuronal homeostasis [157,247,248,250,253,254,255,256]. In microglia, CB2 activation promotes proliferation, migration to lesion sites, and M1-to-M2 phenotype conversion via nuclear factor binding near the κ light-chain gene in B cells (NF-κB). NF-κB inhibition, reducing neuroinflammation [257,258,259,260,261]. Vascular protection includes CB1/CB2-mediated maintenance of blood–brain barrier integrity, inhibition of leukocyte infiltration, β-amyloid clearance, and restoration of cerebral blood flow [262,263,264,265,266]. Collectively, eCBs signaling integrates neuronal, glial, and vascular mechanisms to preserve homeostasis, limit excitotoxicity and inflammation, and promote neuroprotection in ND disorders. ND, neurodegenerative; eCBS, endocannabinoid system; CB1R, cannabinoid type 1 receptor; GABA, gamma-aminobutyric Acid; GPCRs, G protein-coupled receptors; PPAR, proliferator-activated receptor; NF-κB, nuclear factor kappa-light-chain-enhancer of activated B cells; CB1, cannabinoid receptors 1; CB2, cannabinoid receptors 2. ↑, increase; ↓, decrease (Figure was created with BioRender.com).

## Data Availability

Not applicable.

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
