# Peer review of "From Synaptic Plasticity to Neurotoxicity: Endocannabinoid Influence on Addiction and Neurodegeneration"

_ijms, 2025, doi:10.3390/ijms262311632_

Round 1
Reviewer 1 Report
Comments and Suggestions for Authors
This review addresses the involvement of the endocannabinoid system in neurodegenerative disorders and drug addiction. While the manuscript is generally well written and the topic is of interest, it would benefit from a more detailed and structured discussion, especially regarding gene expression, epigenetic mechanisms, and substance-specific effects.
Abstract
- The abstract is currently too general.
- Authors should briefly mention the shared molecular or signaling pathways linking neurodegeneration and addiction (e.g., neuroinflammation, synaptic plasticity, or mitochondrial dysfunction).
Introduction
- Figure 1 is informative and visually well designed. No major revision required here.
Section 2: Cannabinoid Receptor Expression
- Line 76 (CB1/CBR1): Please include detailed expression data for CBR1, specifying brain region-specific expression. For instance, CB1 shows high expression patterns in both mature astrocytes and oligodendrocytes.
- Expression profiles can be obtained from: BrainRNAseq; BrainSpan; GEOSTAT and GTEx repositories. Consider adding bar plots or heatmaps showing CB1 expression across brain regions and developmental stages.
- Lines 102–104 (CB2/CBR2): Apply the same approach for CBR2, emphasizing its expression in glial cells and peripheral tissues, while discussing the controversies regarding its CNS expression.
Section 3: This section currently lacks depth and is largely descriptive. It would benefit from:
I strongly recommend subsectioning Section 3.2 by substance (e.g., nicotine, alcohol, cocaine, opioids), which will help organize the literature more clearly.
A deeper exploration of the epigenetic modulation of the endocannabinoid system in drug addiction. Several studies report DNA methylation or histone modifications affecting cannabinoid receptor expression and behavioral responses: 10.1097/FBP.0000000000000326; https://doi.org/10.1042/NS20220034
Genetic Variants and the ECS
Authors should briefly discuss the contribution of genetic variants in FAAH1 and FAAH2 in modulating endocannabinoid signaling and disease susceptibility. For this purpose, I suggest including the following helpful references: https://doi.org/10.1016/j.gene.2025.149703; https://doi.org/10.1111/acer.13210. This point is particularly relevant when linking ECS to addiction vulnerability and neuropsychiatric manifestations.
Author Response
Reviewer 1
This review addresses the involvement of the endocannabinoid system in neurodegenerative disorders and drug addiction. While the manuscript is generally well written and the topic is of interest, it would benefit from a more detailed and structured discussion, especially regarding gene expression, epigenetic mechanisms, and substance-specific effects.
Responses: We appreciate the reviewer's thoughtful evaluation of our manuscript. We have carefully addressed all suggestions, which have substantially improved the clarity, depth, and overall quality of the review.
- The abstract is currently too general.
Responses: We thank the reviewer for this valuable suggestion. The abstract has been revised to reflect the manuscript's scope better and to incorporate the reviewer's comments. Although there are limited direct studies linking drug- or alcohol-induced neurotoxicity to age-associated neurodegenerative disorders, we have synthesized the available evidence describing drug-induced neurodegenerative changes and their mechanistic overlap with neurodegenerative diseases.
- Authors should briefly mention the shared molecular or signaling pathways linking neurodegeneration and addiction (e.g., neuroinflammation, synaptic plasticity, or mitochondrial dysfunction).
Response: We appreciate this insightful suggestion. The revised abstract now includes a concise description of the shared molecular and signaling pathways—such as neuroinflammation, mitochondrial dysfunction, and impaired synaptic plasticity—linking addiction and neurodegeneration.
- Introduction: Figure 1 is informative and visually well designed. No major revision required here.
Responses: We thank the reviewer for the positive feedback.
- Section 2: Cannabinoid Receptor Expression: Line 76 (CB1/CBR1): Please include detailed expression data for CBR1, specifying brain region-specific expression. For instance, CB1 shows high expression patterns in both mature astrocytes and oligodendrocytes. Expression profiles can be obtained from: BrainRNAseq; BrainSpan; GEOSTAT and GTEx repositories. Consider adding bar plots or heatmaps showing CB1 expression across brain regions and developmental stages. Lines 102–104 (CB2/CBR2): Apply the same approach for CBR2, emphasizing its expression in glial cells and peripheral tissues, while discussing the controversies regarding its CNS expression.
Responses: We appreciate the reviewer's suggestion. However, our introduction is intended to provide a concise overview of the endocannabinoid system (eCBS) without diverting from the review's central focus. Several recent reviews, including ours (PMIDs: 394562299, 31265740) and others (PMIDs: 41098856, 41096816, 40967685, 40886859, 38100799), have comprehensively detailed the anatomical and cellular distribution of CB1 and CB2 receptors in both brain and peripheral tissues. Therefore, we retained a brief and focused discussion to maintain thematic clarity.
- Section 3: This section currently lacks depth and is largely descriptive. It would benefit from: I strongly recommend subsectioning Section 3.2 by substance (e.g., nicotine, alcohol, cocaine, opioids), which will help organize the literature more clearly. A deeper exploration of the epigenetic modulation of the endocannabinoid system in drug addiction. Several studies report DNA methylation or histone modifications affecting cannabinoid receptor expression and behavioral responses: 10.1097/FBP.0000000000000326; https://doi.org/10.1042/NS20220034. Genetic Variants and the ECS-Authors should briefly discuss the contribution of genetic variants in FAAH1and FAAH2 in modulating endocannabinoid signaling and disease susceptibility. For this purpose, I suggest including the following helpful references: https://doi.org/10.1016/j.gene.2025.149703; https://doi.org/10.1111/acer.13210. This point is particularly relevant when linking ECS to addiction vulnerability and neuropsychiatric manifestations.
Responses: We agree that this section is largely descriptive, primarily because there are no direct studies examining how exposure to drugs of abuse or alcohol across developmental stages influences aging-related neurodegenerative pathology. For this reason, we chose not to subdivide the section by individual substances. Instead, we presented them together, highlighting their shared impact on neurodegenerative conditions. In the revised version, we have divided this section to address the reviewer's suggestion.
We also acknowledge the reviewer's point regarding epigenetic and genetic variant studies (e.g., FAAH1 and FAAH2). While these are indeed important for understanding the eCBS's role in psychiatric and addiction-related phenotypes, there is currently a lack of direct evidence linking drug- or alcohol-induced epigenetic changes to neurodegenerative outcomes in preclinical models. We have previously discussed the epigenetic modulation of cannabinoid receptors in detail in our earlier review (PMID: 36358910). Including indirect findings here would be speculative; therefore, we have opted to maintain the current focus on mechanistic overlaps directly supported by existing evidence. However, we have included a short future focus on these issues (lines:392-400 and 479-488) in the revised manuscript.
Reviewer 2 Report
Comments and Suggestions for Authors
see attachment.

Author Response
Reviewer 2
There is an urgent need for the treatment of neurodegenerative diseases and addictive disorders and it is a good idea to consider the endocannabinoid system as a putative target of new drugs for this purpose. Basavarajappa and Subbanna have reviewed this topic which currently is in an early stage. Both authors have published up to 10 relevant papers in the field. The topic is described clearly and illustrated in nice figures.
Responses: We sincerely appreciate the reviewer's thoughtful evaluation of our manuscript and the recognition of its relevance and clarity. We have carefully considered and addressed each comment, and these revisions have substantially improved the accuracy, organization, and overall quality of the manuscript.
Points of criticism:
- With respect to the the reward system in the brain, that part related to the glutamatergic neurone and the D1 MSN should be re-written since one cannot understand the philosophy easily (see below). Lines 89-90: There is a contradiction between those lines (activation or inhibition) and Figure 1 (activation only).
Responses: We thank the reviewer for pointing out this discrepancy. Figure 1 has been corrected to accurately depict both the activation and inhibition of the Erk1/2–CREB pathway, ensuring consistency between the text and the figure.
- Figure 2: The source of this figure (as opposed to the other figures) has not been given. It should read Gi/o (lower case o means "other "). The figure contains three anatomical structures the meaning of which is unclear: mitochondria?, endoplasmic reticulum? The third structure (cell with excentric nucleus) is particularly strange.
Responses: We appreciate this observation. Figure 2 has been revised for clarity, with proper labeling of "G proteins". The figure source has also been provided, and the illustration has been refined to accurately represent CB2 signaling to behavior.
- Figure 4: It should read "glutamatergic "like in the text. Which is the transmitter in the D1 MSN? Is the release of this transmitter in- or decreased?
Responses: We thank the reviewer for this helpful suggestion. Figure 4 has been corrected and clarified. Specifically, 2-AG released from D1 MSNs activates CB1 receptors, thereby inhibiting glutamate release. The revised figure legend now explicitly provides these details.
- 285: AEA).
Responses: Corrected as suggested.
- 286: It is not easy to reconcile that part of the text with the right panel of Figure 4. Moreover, I do not understand the meaning of "both MAGL ".
Responses: We thank the reviewer for this observation. The description and labeling of Figure 4 have been clarified, and the text has been corrected accordingly.
- 300-358: Authors should check whether this part of the text might be condensed to some extent.
Responses: We appreciate the reviewer's suggestion. However, we believe this section provides essential context on neurodegenerative mechanisms and their connection to alcohol-related neurotoxicity. Retaining this content is important to maintain the logical flow and scientific completeness of the review.
- 385: Delete "2-AG-".
Responses: Deleted as suggested.
- 423: Start a new paragraph with "Am-"
Responses: Corrected as suggested.
- 451: Authors should explain that CNR1 is the gene related to the CB1 receptor.
Responses: This clarification has been added to the revised manuscript.
- 380-382: I see a contradiction between the text (CB1R activation and CB2R blockade are positive) and Figure 5 (CB1R activation and CB2R activation are positive).
Responses: We appreciate the reviewer's attention to this point. The apparent discrepancy reflects the differential roles of CB1 and CB2 receptors in alcohol-related versus neurodegenerative conditions. This distinction is now clarified in the revised text and figure legend. Specifically, CB1 and CB2 exhibit opposing effects in alcohol-induced neurotoxicity compared to neurodegeneration-related processes illustrated in Figure 5.
- Figure 5: 1st and 4th box: Glia-Neuron signaling improved; Metabolic support improved; 2nd box: Say "Clearance of aggregates "instead of "Aggregates clearance ". 3rd box: clearance; use the same font for all letters in "maintenance ".
Responses: All suggested corrections have been made. Figure 5 now includes the revised wording and consistent font style throughout.
- 515-516: The list of abbreviations should be arranged in a strictly alphabetical manner. Abbreviations starting with a numeral (e.g., 2-AG) should be placed at the very beginning of the list; abbreviations starting with a Greek letter (e.g., Δ9-THC) should be shifted to its very end. The same abbreviation should be used for each item throughout the text (e.g., Δ9-THC only – and not in addition THC). The list of abbreviations is incomplete, e.g., NFκB is missing.
Responses: We have revised the list of abbreviations to ensure consistency and completeness. All abbreviations used in the main text, figures, and figure legends have been included and arranged in strict alphabetical order. Terms beginning with numerals (e.g., 2-AG) appear at the beginning, and those starting with Greek letters (e.g., Δ9-THC) are placed at the end. Additionally, uniform abbreviation usage (e.g., Δ9-THC only) has been implemented, and missing terms such as NF-κB have been added.
Round 2
Reviewer 2 Report
Comments and Suggestions for Authors
The points raised by this reviewer have not been fully addressed.

Author Response
Reviewer 2
- With respect to the the reward system in the brain, that part related to the glutamatergic neurone and the D1 MSN should be re-written since one cannot understand the philosophy easily (see below). Lines 89-90: There is a contradiction between those lines (activation or inhibition) and Figure 1 (activation only).
Responses: We thank the reviewer for pointing out this discrepancy. Figure 1 has been corrected to accurately depict both the activation and inhibition of the Erk1/2–CREB pathway, ensuring consistency between the text and the figure.
With respect to the the reward system in the brain, that part related to the glutamatergic neurone and the D1 MSN should be re-written since one cannot understand the philosophy easily. --- Has not been done.
Responses: We thank the reviewer for this important comment. We have thoroughly rewritten the paragraph describing glutamatergic signaling and D1-MSN mechanisms within the reward circuitry to improve clarity and conceptual flow. The revised version provides a more accessible and coherent explanation of these interactions, and we hope it now addresses the reviewer's concerns.
- Figure 2: The source of this figure (as opposed to the other figures) has not been given. It should read Gi/o (lower case o means "other "). The figure contains three anatomical structures the meaning of which is unclear: mitochondria?, endoplasmic reticulum? The third structure (cell with excentric nucleus) is particularly strange.
Responses: We appreciate this observation. Figure 2 has been revised for clarity, with proper labeling of "G proteins". The figure source has also been provided, and the illustration has been refined to accurately represent CB2 signaling to behavior.
The arrow between receptor and AC is wrong (blunt-ended line would be correct). The square in the bottom part of the figure is not explained.
Responses: We appreciate the reviewer's careful assessment of the figure. We have corrected the arrow between the receptor and AC to a blunt-ended line, as suggested. In addition, we revised the figure legend to clearly explain the square depicted in the lower portion of the figure. We hope these changes satisfactorily address the reviewer's concerns.
- Figure 4: It should read "glutamatergic "like in the text. Which is the transmitter in the D1 MSN? Is the release of this transmitter in- or decreased?
Responses: We thank the reviewer for this helpful suggestion. Figure 4 has been corrected and clarified. Specifically, 2-AG released from D1 MSNs activates CB1 receptors, thereby inhibiting glutamate release. The revised figure legend now explicitly provides these details.
The type of transmitter and the alteration of its release have not been identified in the lower right neurone (as opposed to the lower left neurone).
Response: We thank the reviewer for pointing out this omission. We have revised Figure 4 to clearly indicate the neurotransmitter type and its release for the lower right neuron, ensuring it is consistent with the information shown for the lower left neuron. We hope this clarifies the circuitry as intended.
- 380-382: I see a contradiction between the text (CB1R activation and CB2R blockade are positive) and Figure 5 (CB1R activation and CB2R activation are positive).
Responses: We appreciate the reviewer's attention to this point. The apparent discrepancy reflects the differential roles of CB1 and CB2 receptors in alcohol-related versus neurodegenerative conditions. This distinction is now clarified in the revised text and figure legend. Specifically, CB1 and CB2 exhibit opposing effects in alcohol-induced neurotoxicity compared to neurodegeneration-related processes illustrated in Figure 5.
My point has not been addressed.
Response: We appreciate the reviewer's comment. We are unclear how to further address this point, as we have already discussed the apparent differences in the effects of CB1 and CB2 receptors on alcohol-related behaviors (original page #380–382) and responded accordingly. To our knowledge, all relevant CB1 and CB2-related studies are included in Figure 5, and we do not see additional information to add on this topic.
- Figure 5: 1st and 4th box: Glia-Neuron signaling improved; Metabolic support improved; 2nd box: Say "Clearance of aggregates "instead of "Aggregates clearance ". 3rd box: clearance; use the same font for all letters in "maintenance ".
Responses: All suggested corrections have been made. Figure 5 now includes the revised wording and consistent font style throughout.
"maintenance "has still not a consistent font. Note that the publisher can change the text but not the body of the figure.
Responses: We have corrected this discrepancy in Figure 5.
- 515-516: The list of abbreviations should be arranged in a strictly alphabetical manner. Abbreviations starting with a numeral (e.g., 2-AG) should be placed at the very beginning of the list; abbreviations starting with a Greek letter (e.g., Δ9-THC) should be shifted to its very end. The same abbreviation should be used for each item throughout the text (e.g., Δ9-THC only – and not in addition THC). The list of abbreviations is incomplete, e.g., NFκB is missing.
Responses: We have revised the list of abbreviations to ensure consistency and completeness. All abbreviations used in the main text, figures, and figure legends have been included and arranged in strict alphabetical order. Terms beginning with numerals (e.g., 2-AG) appear at the beginning, and those starting with Greek letters (e.g., Δ9-THC) are placed at the end. Additionally, uniform abbreviation usage (e.g., Δ9-THC only) has been implemented, and missing terms such as NF-κB have been added.
NFκB has still not been explained.
Responses: We have included this missing abbreviation in the revised manuscript.
Round 3
Reviewer 2 Report
Comments and Suggestions for Authors
Please consider two remaining points:
ad 1. I still have difficulties to understand the right part of Fig. 4. If CB1 receptors lead to a decrease in glutamate release, I would expect that the lower amount of Glu at the facilitatory metabotropic m5Glu receptors will decrease (and not increase) DA release from the D1 MSN neurone and will lead to decreased (and not increased) 2-AG formation.
ad 11. Please look at the last letter of "maintenance" in Fig. 5. The "e" has a much bigger size compared to the preceding letters. As a matter of fact, the size of all letters should be identical.
Author Response
Reviewer 2
ad 1. I still have difficulties to understand the right part of Fig. 4. If CB1 receptors lead to a decrease in glutamate release, I would expect that the lower amount of Glu at the facilitatory metabotropic m5Glu receptors will decrease (and not increase) DA release from the D1 MSN neurone and will lead to decreased (and not increased) 2-AG formation.
Responses: We thank the reviewer for this important comment. We have thoroughly rewritten the paragraph describing glutamatergic signaling and D1-MSN mechanisms within the reward circuitry to improve clarity and conceptual flow. The revised version provides a more accessible and coherent explanation of these interactions, and we hope it now addresses the reviewer's concerns. The revised text is added from lines 241-269:
In addition to regulating DA neuron activity within the VTA, eCBs also act at VTA projection sites in the NAc. Within the NAc, eCBs interact with medium spiny neurons (MSNs) and prefrontal cortical glutamatergic inputs, particularly at prelimbic cortex–NAc synapses [135]. eCB signaling plays a multifaceted role in shaping dopamine transmission and reward-related activity, not by directly inhibiting DA release at D1-MSNs, but by modulating the glutamatergic and GABAergic afferents that regulate MSN excitability. Glutamatergic projections from the prefrontal cortex activate postsynaptic mGluR5 receptors, stimulating diacylglycerol lipase (DAGL) to produce 2-AG. The resulting 2-AG acts retrogradely on presynaptic CB1 receptors to reduce glutamate release, fine-tune excitatory drive onto D1-MSNs, and modulate excitatory transmission [136]. Disruption of this mGluR5–2-AG–CB1R pathway eliminates both natural and drug reward-seeking behaviors [137, 138], and activation of these prefrontal inputs can induce 2-AG–dependent long-term depression (LTD) at NAc glutamatergic synapses [136, 139, 140]. Genetic studies further support this mechanism: conditional deletion of mGluR5 in D1- but not D2-MSNs abolishes 2-AG–dependent LTD and prevents reward-seeking behavior [141], whereas pharmacological enhancement of 2-AG in the NAc restores LTD and reward seeking [136].
Although eCB signaling often increases DA concentrations in the NAc by modulating inhibitory and excitatory inputs to midbrain DA neurons and their terminals, cannabinoid effects remain circuit- and cell-type-specific. Activation of CB1Rs on specific glutamatergic or cholinergic afferents can reduce NAc DA levels [142], underscoring the nuanced, pathway-dependent nature of cannabinoid action. Significantly, eCB signaling does not directly inhibit dopamine release from D1-MSNs—these neurons do not release dopamine—but instead shapes dopaminergic modulation indirectly by regulating presynaptic glutamate and GABA release, as well as forms of synaptic plasticity such as eCB-dependent LTD. Together, these findings extend the classical disinhibition model by demonstrating that eCB–glutamate interactions within the NAc are essential for balancing disinhibition and excitation across VTA–NAc circuits. Through this integrated regulation, both endogenous and exogenous cannabinoids exert robust control over reward processing, reinforcement, and vulnerability to addictive behaviors (Fig. 4).
ad 11. Please look at the last letter of "maintenance" in Fig. 5. The "e" has a much bigger size compared to the preceding letters. As a matter of fact, the size of all letters should be identical.
Responses: All suggested corrections have been made. Figure 5 now includes the consistent font style throughout.